# Disruption of Claudin-Made Tight Junction Barriers by *Clostridium perfringens* Enterotoxin: Insights from Structural Biology

**DOI:** 10.3390/cells11050903

**Published:** 2022-03-05

**Authors:** Chinemerem P. Ogbu, Sourav Roy, Alex J. Vecchio

**Affiliations:** Department of Biochemistry, University of Nebraska-Lincoln, Lincoln, NE 68588, USA; cogbu2@huskers.unl.edu (C.P.O.); souravapr@gmail.com (S.R.)

**Keywords:** claudins, tight junctions, cell/cell interactions, *Clostridium perfringens* enterotoxin, membrane proteins, food poisoning, structural biology

## Abstract

Claudins are a family of integral membrane proteins that enable epithelial cell/cell interactions by localizing to and driving the formation of tight junctions. Via claudin self-assembly within the membranes of adjoining cells, their extracellular domains interact, forming barriers to the paracellular transport of small molecules and ions. The bacterium *Clostridium perfringens* causes prevalent gastrointestinal disorders in mammals by employing an enterotoxin (CpE) that targets claudins. CpE binds to claudins at or near tight junctions in the gut and disrupts their barrier function, potentially by disabling their assembly or via cell signaling means—the mechanism(s) remain unclear. CpE ultimately destroys claudin-expressing cells through the formation of a cytotoxic membrane-penetrating β-barrel pore. Structures obtained by X-ray crystallography of CpE, claudins, and claudins in complex with CpE fragments have provided the structural bases of claudin and CpE functions, revealing potential mechanisms for the CpE-mediated disruption of claudin-made tight junctions. This review highlights current progress in this space—what has been discovered and what remains unknown—toward efforts to elucidate the molecular mechanism of CpE disruption of tight junction barriers. It further underscores the key insights obtained through structure that are being applied to develop CpE-based therapeutics that combat claudin-overexpressing cancers or modulate tight junction barriers.

## 1. Introduction to Tight Junction Barriers

For metazoans, the dense packing of epithelial and endothelial cells helps to compartmentalize tissue-specific functions. Although packing greatly reduces intercellular spacing, spaces nonetheless remain prevalent. Cell junctions act as intercellular bridges, providing adhesion between adjacent cells. Tight junctions are the most apical cell junction, with adherens junctions, desmosomes, and gap junctions residing sub-apically [1,2]. While tight junctions help cells adhere, their primary function is regulating the molecular transport of small molecules, solutes, and ions between cells through their paracellular spaces [3,4]. This molecular gatekeeping maintains tissue homeostasis and can be used to fine-tune the molecular properties of a given tissue or cell type. Tissue-specific functions can, thus, be imparted by tuning the permeability—i.e., leakiness—of tight junctions.

Tight junction structure was first observed through electron microscopy (EM) as membrane fusions or “kissing points” between cells [5]. Freeze-fracture EM (FF-EM) later revealed the detailed structure of tight junctions, wherein they appear as anastomosing networks of strands connected to membrane-embedded plaques [6,7]. Cell biology showed that tight junctions are composed of several families of integral membrane proteins, such as the claudins; the tight junction-associated marvel proteins (TAMPs) occludin, tricellulin, and Marvel-D3; the angulins; and the junctional adhesion molecules (JAMs), which work in concert to direct tight junction form and function in the presence of many scaffolding proteins [4]. The complexity of tight junction strand networks observed by FF-EM does not occur in the absence of claudins, making them the major architects of tight junction structure and function [8,9].

It is now understood that claudins create the barriers or charge- and size-selective pores that tight junctions use to govern paracellular transport and fine-tune molecular homeostasis in tissues [3,4]. To control the formation of barriers or pores, claudins self-assemble both laterally in the plane of one cell’s plasma membrane (cis) and with other claudins on neighboring cells across the paracellular space (trans) [10,11]. Various mechanical and chemical insults can disrupt these claudin self-assemblies, either through intra- or paracellular means, which result in a cornucopia of disease states that all stem from losses in tight junction barrier or pore function [12,13,14,15,16]. The Gram-positive bacterium *Clostridium perfringens* produces an enterotoxin (CpE) that is perhaps the most effective natural molecule at breaking down tight junction barriers. CpE disruption of gut homeostasis causes common antibiotic-associated and foodborne illnesses in domesticated animals [3,17,18].

This review focuses on how the determination of claudin structures by X-ray crystallography have helped to elucidate the structural and functional biology of tight junctions. It further provides an up-to-date account of the structural and functional biology of CpE and its potential mechanism for disrupting tight junction barriers that occurs upon binding to claudins. The review’s aim is to highlight the key structures and their associated insights that have advanced our understanding of claudins, CpE, and claudin/CpE interactions and to reflect on how these insights are being applied in the development of CpE-based therapeutics to treat cancer or tune tight junction barriers.

## 2. Claudins

Tight junctions were discovered in 1963, yet it took 30 more years to identify the first membrane protein that contributes to their formation, occludin [19,20]. In 1998, Tsukita and co-workers performed experiments on occludin and found its reduction did not affect the formation of tight junctions, that tight junctions could be formed in tight junction-less fibroblasts by claudins, and that claudins recruit occludin to tight junctions [19,20]. These pioneering experiments were the first to reveal that claudins are solely responsible for tight junction strand formation and that occludin is an accessory to tight junction function. Although other membrane protein families have since been identified that localize to tight junctions, it remains clear that claudins are the backbone of tight junction form and function [21,22,23,24].

Identification of the first two claudins by Tsukita and co-workers led to subsequent classification of the claudin family by cloning and sequence analyses, which showed that all claudins share a conserved WG/NLWCC motif that defines the fold and predicted that all have four transmembrane helices (TMs) [19,20,25,26,27]. Since then, 27 claudins have been annotated in humans that range in size from 23 to 34 kDa. Each subtype localizes specifically to various organs at distinct plasma membrane domains to impart the tissues they inhabit with unique physiological functions [4,11]. Claudins are classified into three categories based on their barrier properties and sequence similarity. The claudins that form barriers are claudin-1, -3, -5, -11, -14, -18, and -19. Claudin-2, -10, -15, and -17 form paracellular pores, while claudin-4, -6, -9, -12, -13, -16, and 20–27 form barriers or pores depending on their tissue localization, expression levels, or binding partners [4,11]. When focusing on sequence similarity, claudins are classified as “classic” (claudin-1–10, -14, -15, -17, and -19) or “non-classic” (claudin-11, -12, -13, -16, -18, and 20–27) [4,11]. Apart from computational topology predictions, the experimental determination of claudin structure eluded researchers for 15+ years post-discovery.

The first experimental structure of a claudin, that from mouse claudin-15 (mCLDN-15) came in 2014 from Fujiyoshi and co-workers using lipidic cubic phase (LCP) crystallization and X-ray crystallography (Protein Data Bank (PDB) ID: 4P79) [28]. The structure revealed that claudins contained four TMs that were organized in a left-handed bundle (Figure 1A). The clockwise arrangement of the TMs puts the intracellular N- and C-termini in close proximity, with TM2 and TM3 connected through an intracellular loop (ICL). Of significance was the observation that TM3 extended beyond the approximate plasma membrane boundary into paracellular space. Claudin extracellular segments (ECS) form an antiparallel β-sheet composed of five β-strands. Strands β1-β4 compose ECS1, with β3 and β4 connected through disulfide bonding between Cys52 and Cys62. The disulfide bond is part of the WG/NLWCC motif, and its formation is required for claudin function [19,20,25,26,27]. Strand β5 and the extended portion of TM3 compose ECS2. The mCLDN-15 has two extracellular helices (ECH). ECH1 resides between β4 and TM2 and is oriented parallel to the membrane surface, and ECH2 is an extended helix from TM3 that is perpendicular to the membrane plane (Figure 1A). These claudin secondary structural elements are connected by five loops. Loops connect β1 to β2, β3 to β4, and β4 to ECH1. ICL connects TM2 to TM3 and a loop links ECH2 to β5. Although sequence identity varies widely in human claudins, it is presumed that all subtypes have conserved these structural elements. Structural conservation promotes functional and self-assembly overlap in claudins, while sequence diversity permits their unique sub-functionalities, such as whether they form barriers or pores.

Self-assembly of the 27 claudin subtypes are hallmarks of tight junction structural and functional biology. Barriers and pores require both lateral cis and paracellular trans claudin/claudin interactions as well as interactions with scaffolding proteins to anchor tight junctions to the actin cytoskeleton [29]. To add to this complexity, a single subtype may self-associate homomerically or heteromerically with other subtypes in cis (philic) and/or in trans (typic) [10]. Although experimentally derived structures of these assemblies are yet to be determined, several in silico studies have predicted that claudin interactions occur through TM2, TM3, β4, or the two ECHs [30]. Within the mCLDN-15 crystal lattice, linear arrangements of claudins were observed that were hypothesized to resemble cis-interacting claudins in vivo [28]. This finding prompted more complex modelling of the cis and trans assemblies that yield paracellular pores (Figure 1B) [31]. As mCLDN-15 is a pore-forming claudin, it stands to reason that barrier-forming claudins interacting similarly but with subtle conformational changes or residue mutations in their extracellular regions could close the pore, producing a barrier to ions. While this model for claudin-made paracellular pores has some experimental validation, newer models and additional experiments have shown that mCLDN-15 can engage in other unique interaction types [32]. These findings and others indicate that claudins may sample a diverse range of interaction types during self-assembly to achieve the structural complexity necessary for tight junction strand, barrier, and pore formation. All models of claudin interactions demonstrate that the ECSs are intricately intertwined and, thus, critical for claudin self-assembly. Any disruption to ECS structure could, thus, be catastrophic to claudin function (Figure 1B).

**Figure 1 cells-11-00903-f001:**
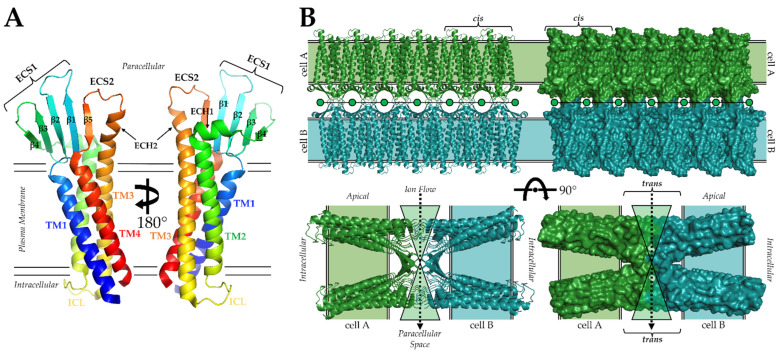
Structure of claudins and model of claudin paracellular pores. (**A**) Structure of representative claudin based on the structures of mCLDN-15 (PDB ID: 4P79) and hCLDN-9 (PDB ID: 6OV2). The protein is colored from N- (blue) to C-terminus (red). (**B**) Model of claudin-made paracellular pores from Suzuki et al. [30]. Claudins from cell A (green) and cell B (teal) interact in trans across the paracellular space, forming a structural hourglass to funnel ions between cells. Cis interactions influence the porosity of tight junctions by governing the number of pores capable of being formed. The aperture of the pore could be closed, forming a barrier, by added steric bulk from amino acid substitutions or slight conformational changes to ECS loops.

## 3. *Clostridium perfringens* Enterotoxin (CpE) and the Identification of Claudins as CpE Receptors

*Clostridium perfringens* is a pathogenic Gram-positive bacterium responsible for both antibiotic-associated and foodborne gastrointestinal diseases in domesticated animals [3,33,34]. Type F strains produce an enterotoxin, CpE, upon sporulation in the animal gut that is required for virulence [3,33,34]. CpE is a 35 kDa protein containing 319 or 325 amino acids. Of these, the CpE gene, with 319 amino acids, is the best known, with the 325 variant only being recently discovered and having unknown clinical relevance [35,36]. In 1997, Katahira et al. determined that CpE bound to two tight junction membrane proteins in humans and mice that were not yet identified as claudins, which they called RVP1 and CpE-R [37]. Later, after the identification of claudin-1 and -2, RVP1 became known as claudin-3, while CpE-R was called claudin-4 [25,26,27]. Importantly, one study determined that CpE receptor capacity, i.e., binding, was lower for claudin-3 than for claudin-4 [27]. The basis for this variance in binding would remain unknown for 24 more years.

The identification of claudins as CpE receptors still left the question of what claudin’s native function was unanswered. In the absence of structure, biochemical evidence had suggested that the C-terminal domain of CpE (cCpE) was responsible for receptor binding activity [38]. Using cCpE, Sonoda et al. discovered that it removed claudins from tight junction strands while reducing tight junction barrier integrity; however, cells did not die [39]. This work with cCpE provided the first evidence that claudins impart barrier function to tight junctions and that cCpE is able to disrupt this function. Later, it was shown with CpE that its incidence in the small intestine also disabled the tight junction barrier and caused morphological damage to gastrointestinal cells and tissues [40]. Taken together, these findings hinted that CpE was a multi-domain protein whose cCpE functioned to bind claudins and whose N-terminal domain (nCpE) functioned to kill epithelial cells in the gut [38].

To decipher claudin recognition of CpE, mutational studies showed that truncations of cCpE’s last 5 to 16 to 30 amino acids resulted in the loss of receptor binding, and, similarly, binding was lost if ECS2 from a receptor claudin was substituted with ECS2 from a non-receptor claudin [41,42,43,44]. At this time, it was experimentally validated that CpE had two domains with distinct functions: cCpE that binds claudins and nCpE that was responsible for cytotoxicity [42]. Experiments to elucidate the role of cCpE’s terminal residues in binding to claudins found that tyrosines 306, 310, and 312 and Leu315 were critical for binding to claudin-4 and that Tyr306 and Leu315 mutants to alanine reduced cCpE’s ability to disrupt tight junction barrier function [45,46,47]. In total, these works revealed several claudin-binding residues of cCpE and that both CpE domains were required to coordinate in CpE function.

Independently, cCpE cannot kill cells and nCpE cannot bind cell receptors. So, how does nCpE induce cytotoxicity? The current model for CpE-induced cytotoxicity begins with 35 kDa CpE binding to 25 kDa receptor claudins via cCpE, forming a ~60 kDa “small” complex [48]. Intact “small” complexes oligomerize as hexamers to form a ~360 kDa pre-pore complex [49,50]. Depending on the cell line or tissue, non-receptor claudins and/or occludin may associate with pre-pore complexes, forming larger complexes of 400–600 kDa [48,51,52]. Once large complexes are formed, the nCpE portion is thought to undergo a structural rearrangement to form the cytotoxic membrane-penetrating β-pore, while cCpE remains bound to claudins to tether pores near the plasma membrane. The residues responsible for β-pore formation reside between amino acids 81 to 106, and pores are highly selective for cations, such as Ca^2+^ [50,53,54,55,56]. Calcium influx activates oncosis or apoptosis signaling cascades depending on the amount of CpE present, killing individual epithelial cells [55]. It is believed that CpE binding to claudins may occur outside of tight junctions to surface-exposed claudins, as the application of cCpE to only the basolateral compartments affects tight junction barrier integrity [39,57]. Thus, CpE disabling tight junctions may occur indirectly as a result of first inducing damage to cell surfaces, which then exposes tight junction-associated claudin receptors, forming new β-pores subapically [58]. Yet, whether CpE binding occurs outside tight junctions to prevent claudin assembly or it breaks intact tight junctions through disruptions to claudin cis and/or trans interactions has not been unequivocally determined.

## 4. Structural Biology of cCpE and CpE

The first structural information of CpE came in 2008 from Van Itallie et al. when they determined the X-ray structure of cCpE at 1.75 Å (PDB ID: 2QUO) [59]. The 14 kDa domain revealed a 9-stranded β-sheet that exhibited structural similarity to other spore-forming bacterial toxins (Figure 2A). This was unexpected because of the absence of sequence homology between these proteins. In addition to deciphering the cCpE fold, the structure revealed that cCpE’s terminal 30 amino acids formed a pocket near β8 and β9 and that tyrosines 306, 310, and 312 reside in close 3D proximity, supporting the results from previous studies that showed this region and its residues were important for claudin binding [41,42,44].

In 2011, further insights were gained into CpE function with the determination of CpE structures by Kitadokoro et al. (PDB ID: 3AM2) and Briggs et al. (PDB IDs: 2XH6 and 2YHJ) [60,61]. The CpE monomer is composed of 17 β-strands and 2 to 3 α-helices (Figure 2B). These structures revealed the two-domain architecture of CpE, where the cCpE portion agreed with the structure from Van Itallie et al., and the nCpE structure was structurally homologous to monomers of the other β-pore-forming toxins, such as *C. perfringens* ε-toxin and aerolysin. To support the role of the N-terminal domain in oligomerization and pore formation, the CpE structures show that a purported pore-forming helix present in nCpE can undergo conformational change without affecting the overall fold of the rest of the molecule. Moreover, a serine and threonine surface track, which is present in aerolysin and is responsible for oligomerization and membrane orientation, is also present in CpE [62,63]. In the three CpE structures, the first 34 to 35 residues were not observed in the crystal, indicating they were either proteolytically removed or disordered. N-terminal deletion or removal by trypsin results in two- to three-fold increased cytotoxicity, so these residues are not required for CpE function [42,64]. Overall, these first CpE structures suggested that residues comprising Ser93 to Phe103 constitute the pore-forming helix and that the helix must undergo a secondary structural transition to form a β-barrel pore. CpE trimers were crystallized or generated through crystallographic symmetry and are formed by interactions between acidic side chains, such as Asp48, as well as interactions with divalent cations [61]. The trimers are likely not functional, and mutation to the acidic side chains also does not alter CpE cytotoxicity [60]. Consequently, Briggs et al. proposed a hexameric model for the CpE β-pore that was adapted from a homologous toxin that forms a dimer of trimers [61]. This model showed that Asp48 would lie near dimer interfaces, and that the pore-forming helix and claudin-binding pocket of cCpE would reside on the same side of the hexamer (Figure 2C). This model likely represents a pre-pore, as the membrane-spanning β-barrel pore is not present.

**Figure 2 cells-11-00903-f002:**
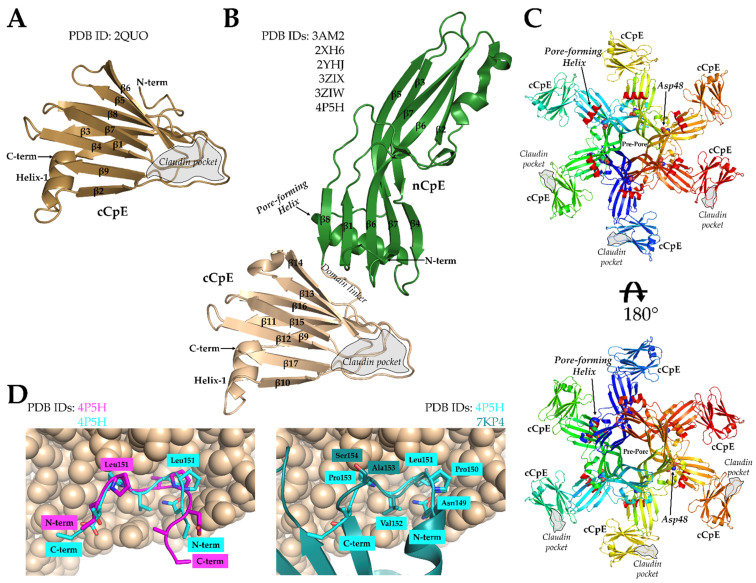
Structures of cCpE and CpE and a model of the CpE pore. (**A**) Structure of cCpE (brown), PDB ID: 2QUO from Van Itallie et al. [59]. (**B**) Consensus structure of CpE from Kitadokoro et al., Briggs et al., and Yelland et al. [60,61,65]. The nCpE (green) and cCpE (brown) domains are shown as cartoons. (**C**) The model of a CpE pre-pore based on the structure of hemagglutinin HA70/HA3 from Clostridium botulinum as proposed by Briggs et al. [61]. Each monomer is colored independently. The Asp48 in each subunit is shown as spheres and the pore-forming helix (red) as a cartoon. The pore-forming helix faces the membrane when cCpE is bound to claudins. (**D**) The peptide from claudin-2 (magenta) bound to the cCpE domain of CpE (brown, spheres) from PDB ID: 4P5H by Yelland et al. [65]. Our re-refined orientation of the claudin-2 peptide (cyan) compared to the original structure (left) and to the ECS2 of claudin-4 from PDB ID: 7KP4 (right; teal). The re-refinement shows the peptide can fit the electron density in both poses, but the pose that resembles 7KP4 makes better structural sense considering recent publications.

The next insights came from Yelland et al., who determined the structures of CpE with the first 37 residues at the N-terminal deleted (PDB ID: 3ZIX), an Asp48Ala mutant (PDB ID: 3ZIW), and CpE bound to a peptide from the ECS2 of claudin-2 (PDB ID: 4P5H) [65]. As CpE structures determined previously had disordered N-termini, the N-terminal deleted structure showed no major structural variance. Moreover, while the Asp48Ala mutant of CpE was shown to be biochemically unable to form pre-pore complexes and, thus, to be cytoxically inert, the structure of CpE Asp48Ala showed no major differences from the wild type, indicating that this functional inactivity was not structurally related [50,65]. The last crystal structure had a peptide from mouse claudin-2′s ECS2 (^141^HGILRDFYNPLVPDAMKFEI^160^) bound to CpE. Because claudin-2 does not bind CpE naturally, this peptide was modified to include an NPLVP^151^ motif (underlined) from mouse claudin-3 to impart toxin binding, as was determined by Winkler et al. [57]. The binding of this peptide did not alter the CpE structure when compared to non-peptide-bound CpEs. The peptide bound in a groove in cCpE between β8 and β9 as predicted previously [59]. However, the peptide’s orientation in the claudin-binding pocket, N- to C-term, was opposite to those from prior modelling studies (Figure 2D). In studies by Veshnyakova et al. using in silico and in vitro methods, cCpE was predicted to bind the claudin ECS2 loop C- to N-term [66]. The crystal structures determined later of cCpE in complex with claudins would reveal that the claudin-2 peptide was indeed bound backward and that it bound C- to N-term as predicted by Veshnyakova et al. This result stems from either truly binding backward as a function of being a peptide and not a fully folded protein or from model building ambiguity in the 3.4 Å resolution electron density map. Using the Electron Density Server map for PDB ID 4P5H, we rotated this peptide 180° then real-space refined it into the experimental electron density map using Coot [67,68]. We found an orientation with good stereochemistry that fits into the map and resembles the native ECS2 of claudin-4 when in complex with cCpE (Figure 2D; PDB ID: 7KP4) [69]. The differences that exist between the modified claudin-2 peptide and the native claudin-4 ECS2 are a result of the peptide containing Pro150 and Pro153, whereas claudin-4 has a Pro150 and Ala153.

In sum, these findings illuminated the structures of cCpE, CpE, a potential CpE oligomer, and how CpE may bind claudins. Moreover, they provided the structural basis of and validation for numerous functional experiments concluded previously that were incompletely understood. Collectively, it became clear that CpE binds claudin receptors using cCpE and that the process of forming cytotoxic pores requires sequential steps involving oligomerization and structural rearrangements of CpE. Finally, these works provided a foundation for elucidating the amino acid determinants of CpE recognition of claudins, which, through biochemical analyses, had yielded ambiguous findings up to that point. The structures of claudin/enterotoxin complexes would be essential to clarify these ambiguous functional results.

## 5. Structures of Claudins in Complex with cCpE

The classification of claudins that are receptor or non-receptors for CpE was being determined using biochemical, biophysical, or cell-based assays that qualified cytotoxic effects. Veshnyakova et al. provided an exhaustive account of the methods and findings from work conducted prior to 2010 [18]. In short, these efforts showed that claudin-3, -4, -6, -7, and -9 were high-affinity receptors and claudin-1, -2, -5, -8, -14, and -19 were low-affinity receptors, while all other claudins were true non-receptors and do not bind cCpE [18,39,43,57,70,71,72,73]. As mentioned previously, however, ambiguous and sometimes contradictory findings were obtained due to differences in the claudin orthologs or cCpE or CpE constructs and the amounts of each used in the assay as well as the assay’s sensitivity. Specifically, claudins -1, -2, and -5 were found to be both insensitive or sensitive to CpE toxicity [37,39,57,70,72]; claudin-4 and -19 were found to bind or not bind CpE [37,57,74]; there was disparity in claudin-3 and claudin-4 affinities for CpE [37,75,76]; and there was discord in what interaction types and residues direct enterotoxin binding [57,70]. To confound progress, claudin subtypes from different species (orthologs), such as claudin-4, did not bind enterotoxin identically [57]. Even the types of interactions enterotoxins form with claudins were difficult to pinpoint, as it was found that both electrostatic and hydrophobic interactions via the NPLVA^153^ motif could influence cCpE binding [57,66,70]. Prior to structures, a comprehensive understanding of CpE receptors and the details of their interactions remained elusive.

In 2015, Fujiyoshi and co-workers determined the second structure of a claudin and the first of a claudin in complex with cCpE, that of mouse claudin-19 (mCLDN-19) at 3.7 Å (Figure 3A; PDB ID: 3X29) [74]. To facilitate crystallization, 26 residues were removed from the C-terminus of mCLDN-19 and 3 membrane-proximal cysteines were mutated to alanines to remove heterogeneous palmitoylation. To improve diffraction, a Ser313Ala mutant was introduced to cCpE. In the crystal’s asymmetric unit, two antiparallel mCLDN-19/cCpE complexes associated head-to-toe in 1:1 stoichiometry. The overall structure of mCLDN-19 resembled mCLDN-15, and the bound cCpE was structurally homologous to the cCpE structure determined in the absence of claudin [28,59]. Two notable structural differences were found between mCLDN-15 and -19, however. Whereas the structure of mCLDN-15 resolved ECH1 but not the loop connecting β1 to β2, mCLDN-19 possessed a β1-β2 loop but no ECH1. These structural differences were purported to arise from cCpE effects to the claudin structure for mCLDN-19, as mCLDN-15 was not bound to the enterotoxin. Importantly, the structure revealed the ECS2 portion of mCLDN-19, which contains an NPSTP^153^ motif, bound in the proposed claudin-binding groove near β8, β9, and tyrosines 306, 310, and 312 of cCpE. Moreover, the structure showed that both ECS1 and ECS2 coordinate to bind to cCpE, which was surprising given the biochemical data that suggested only ECS2 was involved in binding [43,70]. This work concluded with hypotheses into the structural basis and mechanism of cCpE-mediated disruption of tight junctions, which we will summarize in the ensuing sections of this review.

The next crystal structure of a claudin/cCpE complex came from Shinoda et al. in 2016 using human claudin-4 (hCLDN-4), which was resolved to 3.5 Å (Figure 3B; PDB ID: 5B2G) [77]. Again, because crystallization and diffraction of small membrane proteins is challenging, a soluble protein fusion containing T4 phage lysozyme (T4L) was engineered at the N-terminus of hCLDN-4, and the C-terminus was truncated by 26 residues. The T4L induced the proteins to pack tightly within the crystal as four antiparallel hCLDN-4/cCpE complexes in 1:1 stoichiometry with the T4L of one complex capping the cCpE from another. The overall structure confirmed that both ECSs of claudins interact with cCpE. In fact, the mutation of ECS1 residues showed losses to binding, verifying this result. However, because hCLDN-4 is a high-affinity CpE receptor and mCLDN-19 is not, structural differences arose. Specifically, Leu151 of hCLDN-4′s NPLVA^153^ motif was found to deeply access the triple tyrosine claudin-binding pocket of cCpE. This finding agreed with previous biochemical work that showed this motif played a vital role in cCpE binding [18,45,57,66]. The authors also observed conformational differences in ECH2, which bent more in hCLDN-4, and that cCpE amino acid interactions with hCLDN-4 or mCLDN-19 varied. The Phe35Tyr and Leu151Ser mutations from hCLDN-4 to mCLDN-19 were thought to drive these differences, and the Ser313Ala mutant of cCpE that is required for the crystallization of mCLDN-19 likely makes this non-receptor more receptive to CpE. Another important contribution of Shinoda et al. was quantifying the affinity of cCpE for hCLDN-4, which they showed bound with a K_D_ of 3.4 nM using surface plasmon resonance [78]. Previously, these affinities were quantified using indirect means [43,70,74,79]. Using their structure and a homology model of hCLDN-4 not bound to cCpE as guides, Shinoda et al. hypothesized what the mechanism of tight junction disruption triggered by cCpE may be. We will present this in Section 6.

In 2019, Fujiyoshi and co-workers determined three crystal structures of another CpE receptor, mouse claudin-3 (mCLDN-3), in complex with Ser313Ala cCpE [79]. The structures consisted of Ser313Ala cCpE bound to wild type (PDB ID: 6AKE), Pro134Gly (PDB ID: 6AKG), and Pro134Ala (PDB ID: 6AKF) mCLDN-3; and diffracted X-ray to 3.6, 4.3, and 3.9 Å, respectively (Figure 3C). Like mCLDN-19, 36 residues from the C-terminus and alanines that produced de-palmitoylated protein were used for mCLDN-3 crystallization, and two antiparallel mCLDN-3/cCpE complexes associated head-to-toe in 1:1 stoichiometry in the asymmetric unit. The Ser313Ala cCpE mutant was found to increase complex thermostability and improve diffraction from 8 to 4 Å [80]. Although the individual structures of mCLDN-3 and cCpE were like other claudin/cCpE complexes, the binding mode of cCpE differed significantly. Typically, the cCpE interactions with the extracellular surface of claudins produce an intimate interface with minimal solvent-accessible area. In these structures, it was observed that the interfaces were so large and exposed to solvent that a true interface did not exist. The mCLDN-3 bound cCpE using only its loop connecting β1 to β2 and its NPLVP^151^ motif. While the structures provided new insights into claudin/cCpE interactions, the paper focused on elucidating the mechanism of tight junction strand morphology. Nakamura et al. compared all then-existing crystal structures of claudins and observed that ECH2 is bent outward at various angles while the TM3 portion remains similar within the membrane. The authors attribute these angular differences to the presence or absence of Pro134. In mammalian claudins, they propose, Pro134 would bend ECH2, while Gly134 or Ala134 would yield ECH2 that were linear from TM3. Observing cells transfected with mutants of position 134 using FF-EM, they observed that Pro134-containing claudins make straight tight junction strands, while alanine- or glycine-containing claudins yield curvy strands. They hypothesize that strand morphology in native epithelia is driven by the constituent claudins and the angle of their respective ECH2, which yield either straight or curved strands. We made similar comparisons using structures available from the PDB and used non-crystallographic complexes from the same asymmetric units to explain these differences in the context of cCpE binding (Figure 3D). The ECH2 portion of TM3 does vary between structures, but the slight differences are likely within structural error and/or caused by cCpE movements, which, in turn, alter the conformations of claudin ECS. ECH2 movements in other Pro134-containing claudins, such as hCLDN-4 and -9, are not as sizeable as mCLDN-3, indicating that the ECH2 orientation observed for mCLDN-3 may be a result of a cCpEs binding mode that lacks claudin/cCpE interactions at its interface rather than the presence or absence of Pro134. In vivo results appear to validate that Pro134 and ECH2 alter tight junction strand morphology, however.

Our laboratory’s contributions began in 2019 with two crystal structures of cCpE in complex with human claudin-9 (hCLDN-9) at 3.2–3.3 Å (Figure 3E) [81]. The two structures were similar, apart from the size of the solvent-accessible pockets at the hCLDN-9/cCpE interface. The binding mode of cCpE varied by 15°, resulting in one being more closed (PDB ID: 6OV2) and the other more open (PDB ID: 6OV3) to solvent. Unlike previous structures of claudin/cCpE complexes, neither protein contained truncations or mutations. The presence of an intact C-terminus, though not visible, likely prevented antiparallel packing of two complexes, making these structures the first with single copies of 1:1 stoichiometric claudin/cCpE complexes in their asymmetric units. This crystal packing eliminated steric influence on the binding mode and interactions between hCLDN-9 and cCpE from other complexes. These were also the highest resolution structures. Prior to discussing cCpE interactions, we used the structure of hCLDN-9 to investigate functional data on this poorly understood claudin. Previous work by Nakano et al. and Sas et al. established that claudin-9 forms a barrier to cations and is a likely an anion-selective pore-forming claudin, and that a Phe35Leu mutant causes deafness by cation influx in the inner ear [82,83] The structure showed where Lys65 and other paracellular pore-lining residues responsible for anion selectivity in the inner ear reside and explained how the Phe35Leu could cause deafness through the destabilization of cation barriers and where Val32 and Val35 reside and how they may be used by the hepatitis C virus for cell entry [82,84]. To characterize the toxin, we first measured the affinity of cCpE and CpE for hCLDN-9 using bio-layer interferometry (BLI) and found they bound with K_D’s_ ~5 nM, similar to hCLDN-4, confirming it as a high-affinity CpE receptor [78]. Although the hCLDN-9/cCpE interactions varied between the closed and open conformations, both structures retained tight interactions between the NPLVA^153^ motif of hCLDN-9 and the claudin-binding pocket of cCpE. An analysis of interface residues showed that Leu151 had the greatest influence on complex formation. The role of Leu151 was verified by data from cCpE binding to a Ser151Leu mutant of mCLDN-19, which showed a 20-fold increase in affinity [74]. The remainder of this paper describes our proposed mechanism for cCpE-based disruption to claudin assemblies and, thus, tight junction barriers using information from all previous structures and these new structures, which we will expand on in Section 6.

Our latest contribution, in 2021, resolved a new crystal structure of cCpE in complex with hCLDN-4 to 3.4 Å (Figure 3F; PDB ID: 7KP4) [69]. Like hCLDN-9, the proteins contained no alterations and packed in 1:1 stoichiometry as a single complex. The novel crystal packing and the complex not requiring T4L for crystallization proved beneficial and allowed the hCLDN-4/cCpE complex to reach an energy-minimized form unperturbed. In the structure, we observed standard features, such as both ECS coordinating cCpE binding and the NPLVA^153^ motif penetrating the cCpE claudin-binding pocket. However, we also observed new features of the claudin/cCpE interface that did not exist in other structures or went unnoticed. For instance, we discovered a 12 amino-acid-containing “cCpE-binding motif” formed via an intermolecular hydrogen bonding network that extends from the membrane surface to the NPLVA^153^ motif. The core of the cCpE-binding motif is Arg158, which stacks in a three-arginine string within the solvent-accessible interface. A sequence analyses of the cCpE-binding motif showed that it has conservation in claudins but varies at the third position of the NPLVA^153^ motif, Leu151. This led us to quantify and compare enterotoxin binding to various claudin subtypes from humans and mice using BLI. We found that claudins with a Leu at position three bound enterotoxins with a higher affinity than those with other hydrophobic side chains, which we confirmed with point mutants and chimeras. Interestingly, we showed that human claudin-3 (hCLDN-3) had a 100-fold lower affinity and human claudin-1 (hCLDN-1) had a 380-fold lower affinity for cCpE compared to hCLDN-4. The discovery of the cCpE-binding motif explained results from Katahira et al. in 1997 that showed claudin-3 had a lower receptor capacity for CpE than claudin-4. Although our results showed greater disparity between enterotoxin binding to hCLDN-3 vs. hCLDN-4 than Katahira et al., we attribute this to numerous differences between analyses. Specifically, claudins in tissues/cells vs. in solution, variable receptor:enterotoxin ratios, the presence of endogenous claudins during analysis vs. samples with single pure claudins, and steady-state vs. pre-steady-state kinetic determinations. Similar analyses to Katahira et al. showed larger, up to 10-fold differences in low- vs. high-affinity receptor binding to CpE, indicating that tissue/cell/claudin subtype differences are challenging to control [75,76]. The existence of a cCpE-binding motif is another piece of evidence that helps to link the previous findings that electrostatics and NPLVA^153^-driven interactions govern cCpE binding to claudins [27,43,57,70,71]. Mutagenesis of other residues of the motif will shed further light. Lastly, we postulated that hCLDN-1 and -3 are not viable receptors for CpE at pathophysiologically relevant concentrations, which range from 3 to 350 nM [71]. Therefore, we hypothesized the native receptor for CpE in humans is hCLDN-4, while in mice it is mCLDN-3 and potentially -8.

These structures of cCpE bound to claudins rapidly accelerated our understanding of CpE structural and functional biology in a short period of time. Table 1 provides a summary of what was detailed in Section 5. In aggregate, findings produced by these structures have helped to clarify previously generated functional results that were often ambiguous, providing the structural basis for the formation of the ~60 kDa “small” complex that precedes CpE oligomerization and cytotoxic pore formation. Questions remain as to whether CpE targets claudins outside of tight junctions to disable the tight junction barrier or if it targets and directly disrupts intact claudin assemblies at tight junctions to do so. The structures of claudin/cCpE complexes offered important new insights that generated testable hypotheses, some of which still require experimental validation.

## 6. Mechanisms of CpE Disruption of Claudins and Tight Junctions

To understand CpE-induced disabling of claudin assemblies that lead to disruptions to tight junctions, we must first discuss how we believe claudins assemble at the sub-molecular level. One of the first models for claudin cis assembly came from the observation by Suzuki et al. that mCLDN-15 in LCP crystals packed in linear arrangements [28]. In crystals, claudin/claudin interactions were driven by the Met68 in one protomer binding within a hydrophobic cavity created by Phe 146, Phe 147, and Leu 158 of the adjacent protomer (cis1). Mutations of these residues to smaller or charged side chains disrupted tight junction strand formation on insect cell plasma membranes as assessed by FF-EM, giving in vivo validation that LCP linear arrangements may be physiological. Suzuki et al. would use this structure-based model as the impetus to model a larger complex of cis- and trans-interacting mCLDN-15 that form paracellular pores (Figure 1B) [30]. Here, the Asn61 in β4 is also involved in the cis interface and with the other residues, arranging claudins in the linear antiparallel double-rows. They propose that cis-assembled double-rows from one cell could interact with similar complexes on adjoining cells to form a “β-sheet channel” in paracellular space (Figure 1B). Another model by Zhao et al. suggested that other cis interfaces were possible for mCLDN-15 [31]. In this model, which was derived computationally then tested in silico and in vivo, the interface consists of residues Ser67, Arg79, Phe146, Phe147, Leu158, and Glu157 (cis2). Cis1 and cis2 are distinct but overlap partially in the residues involved. Using fluorescence microscopy and FF-EM in mammalian cells, Zhao et al. confirmed that Phe146 and Phe147 mutants abolished tight junction strand formation but also that Met68 mutants did not. Similar assays showed Ser67, Arg79, and Glu157 mutants altered strand morphology and/or continuity in vivo, indicating that these residues are involved in later cis assembly between neighboring claudins. Later, Hempel et al. further developed models to include not just cis1, the face-to-face double-row model proposed by Suzuki et al., but added a new antiparallel back-to-back arrangement (cis3) [32]. Cis3 differed from other cis models in that this model was optimized for the tight packing of TM2 and TM3. Mutagenesis of claudin/claudin interfaces was then tested using cysteine cross-linking in tight junction-less HEK293 cells and was analyzed by confocal laser scanning microscopy and FF-EM. In short, all mutants formed strands, but Phe34, Ile39, Leu69, and Leu70 mutants altered strand formation more than other residues; the in vivo and in vitro data suggested the back-to-back cis3 assembly was not viable; and the cis1 model interactions were consistently reproducible, while the cis3 interactions were not. Owing to differences in interactions within and between different claudin subtypes as well as cis oligomerization outside cell/cell contacts versus cis polymerization at cell/cell contacts, this work helped to clarify these complexities and surmised that extracellular cis1 and trans interfaces mutually facilitate barrier or pore formation by either electrostatic attraction or repulsion between claudin ECS elements, respectively. In sum, these works established models for the claudin cis and/or trans interactions necessary for tight junction barrier and pore functions that are paramount for deciphering the mechanisms of CpE-induced tight junction disruption.

These models of cis and trans claudin assembly help us understand the normal structure and function of tight junctions (Figure 4A). During *Clostridium perfringens* pathogenicity, McClane and co-workers proposed a mechanism of CpE-mediated disruption of tight junction structure and function (Figure 4B) [58,85]. This process entails CpE binding to exposed claudin monomers or cis oligomers at apical surfaces then inducing damage to cells via β-pore formation. Morphological damage caused by β-pore-mediated Ca^2+^ influx subsequently induces cell death signaling through either oncosis or apoptotic processes, disrupting tight junctions by degrading claudins. Opening the tight junction would then expose basolateral claudins to CpE binding and cells to further insult. In this mechanism, morphological damage to cells at their apical domains would indirectly dissociate claudin assemblies to disrupt tight junctions. Experimental support for this mechanism came from Eichner et al., who showed that CpE-induced damage to colon epithelia required apical claudin receptors and that tissue damage was limited by the maintenance of partial tight junction integrity, which prevents CpE from full access to basolateral claudins [86]. Disruption of the tight junction barrier via this indirect mechanism would not require direct disabling or disruptions to claudin assemblies by CpE. The structures of the claudin/cCpE complexes determined previously would, thus, do little to prove or disprove this mechanism of tight junction disruption by CpE, apart from revealing the interactions necessary for CpE recognition of claudin receptors and how cCpE binds claudins during the formation of the “small” complex.

Clearly and foremost what the structural biology of the claudin/cCpE complexes revealed was that trans interactions between claudins on neighboring cells could not exist if claudin was bound to cCpE (Figure 4B). No structural rearrangements or loss to side chain interactions are necessary to explain this, because all structures showed that cCpE fully occupies both ECS. Thus, steric hindrance by cCpE or CpE would abrogate any potential trans interactions. Occupying interfaces used for claudin/claudin trans interactions is, therefore, another, more direct, mechanism by which cCpE may disrupt tight junction barriers. In the context of tight junction strand formation, however, this mechanism could disable tight junction function only by binding monomeric or cis-oligomerized claudins outside of cell/cell contacts where they are not polymerized and trans assembled. Sonoda et al. speculated that this “sequestering” mechanism could function by cCpE occupying non-polymerized pools of claudins and preventing their integration into polymerized strands if tight junctions are indeed maintained by such an equilibrium [39]. Their observation that cCpE only affected tight junction barrier integrity when applied to the basolateral and not apical compartments appears to validate this mechanism [39,57]. Moreover, recent findings suggest that claudins are indeed added to tight junctions through strand breaks at their basal edges [87]. Here, cells would not be in contact, and, thus, CpE could bind non-polymerized claudins to prevent their trans assembly and integration into strands via this “sequestering” mechanism. The finding that only a basal application of cCpE disrupts tight junctions hints that the enterotoxin cannot directly break intact and polymerized junctions at cell/cell contacts. However, in the gut, no explanation for how CpE could gain access to the basolateral compartments to initiate sequestration has been offered. To do so it must first disrupt apical tight junctions. The disabling of claudin/claudin trans interactions through sequestering may very well be the only direct mechanism of cCpE-induced disruption of tight junctions; whether CpE functions similarly remains an open question but the steric occlusion of trans interactions by CpE may occur if cCpE can do so. To validate sequestering, further experiments are required to understand how or whether CpE gains access to basal compartments or if non-polymerized pools of claudins are synthesized apically, polymerized, and are then integrated into tight junction strands. In colon epithelia, CpE has been shown to target apically presented non-polymerized claudin receptors to invoke cell damage [86].

Because obstruction of claudin/claudin trans interactions appeared obvious to all researchers that determined claudin/cCpE structures, each research group proposed mechanisms for tight junction disruption by CpE that focused on the cCpE-induced disabling of claudin cis assemblies [74,77,81]. For these hypotheses, the structure of mCLDN-15 from LCP proved vital because it was not complexed with cCpE, and, thus, structural comparisons could be made to decipher the conformational effects of cCpE binding to the claudin structure. Generally, all claudin/cCpE structures revealed the absence of ECH1 and the conformational flexibility of ECH2 in claudins when bound to cCpE, as compared to apo mCLDN-15. These two helices house many of the side chains involved in mCLDN-15 cis assembly: Ser67, Met68, Leu69, Leu71, Phe146, and Phe147. If these helices are used in cis assembly, cCpE binding would structurally perturb them and their associated side chain orientations, thus disrupting inter-claudin cis interactions. The perturbations to cis-interacting residues caused by cCpE, these researchers would show, could affect both cis1 and cis2 models for assembly.

Saitoh et al. used the mCLDN-15 structure to make a homology model of mCLDN-19 in its apo form, then compared that structure to their mCLDN-19/cCpE complex [74]. They found that disordering ECH1 could disrupt the linear cis arrangements of claudins based on the LCP packing of mCLDN-15 that yielded model cis1 [28]. Specifically, Leu70, Phe148, and Phe149 on ECH1 and ECH2 in mCLDN-19 would cease to form hydrophobic interactions with neighboring protomers and, thus, destabilize linear polymeric claudins in tight junction strands. Shinoda et al. used a similar approach but provided a new insight [77]. They first made a homology model of apo hCLDN-4 based on mCLDN-15 and compared it to their hCLDN-4/cCpE structure. Apo hCLDN-4 would form intramolecular interactions between Tyr67, Leu71, and Leu77, while cCpE-bound hCLDN-4 would not, due to an unwinding of the ECH1 helix from the displacement of Tyr67 upon cCpE binding. Their structural analyses came to similar conclusions as Saitoh et al., showing that if hCLDN-4 assembled in cis1 arrangements, cCpE binding would disturb ECH1/ECH2 contact interfaces. These two structures of mCLDN-19 and hCLDN-4 bound to cCpE proposed a potential new mechanism for cCpE-induced disruption to tight junctions by the disabling of cis interactions, which still awaits experimental support. Shinoda et al. extended their analysis to show that if hCLDN-4 formed linear cis1 arrangements, a cCpE would collide with cCpEs from adjacent claudin/cCpE complexes. The cis1 assemblies proposed by Suzuki et al. based on the LCP crystal packing in mCLDN-15 appear to pack so tightly that cCpE would have difficultly accessing its complementary claudin surface [28,30,77]. This unavoidable steric hindrance caused by neighboring cCpEs would, thus, disfavor 1:1 claudin/cCpE stoichiometric complexation due to spatial constraining, if in fact cis1 arrangements form in vivo.

Because the structures of mCLDN-19 and hCLDN-4 in complex with cCpE provided similar mechanisms for the cCpE-induced disabling of claudin cis assemblies, we theorized how cCpE could disrupt other potential cis interfaces and amino acids using the model of cis2 [31,81]. We made a model of a homodimeric cis2-interacting hCLDN-9 bound to cCpE and compared it to an apo homodimeric cis2-interacting mCLDN-15 to assess how cCpE binding could alter this interface or the conformations of amino acids used there. When assembled in cis2, these models showed that of the five polar or non-polar side chain interactions between the apo mCLDN-15 or hCLDN-9 dimers, cCpE binding would result in a loss of four to five of these interactions. Similar results were found in the hCLDN-4/cCpE structure [69]. Ser67, Leu71, Gln76, Arg79, Phe146, Phe147, and Glu157 (mCLDN-15 numbering) would be specifically affected. Mutants of these residues were found by Zhao et al. to alter tight junction strand formation or morphology in vivo [31]. Furthermore, the upward rocking movement of cCpE observed in the two hCLDN-9 “closed” to “open” structures showed that cCpE binding alters the area where ECH1 normally forms as well as the side chain orientations on ECH2. This glimpse into possible cCpE-induced dynamics was likely to cause added abrogation of the cis2 interface. Importantly, when we took the homodimeric cis2-interacting hCLDN-9 model and further modelled CpE onto the experimentally determined structure of cCpE, it revealed that neither domain of CpE would clash significantly with the adjacent CpE molecules. This exercise showed that all perturbations to claudin structure and cis assembly would be induced exclusively by the enterotoxin and not steric hindrance. This is a distinction between the cis1- and cis2-based models for the cCpE-induced disabling of claudin cis assembly. Because the cis2 assembly proposed by Zhao et al. suggests that CpE will not sterically clash if engaged 1:1 with claudins, if both cis1 and cis2 claudin assemblies are present in vivo, CpE could preferentially bind to cis2-assembled claudins due to unconstrained access to its claudin-binding surfaces.

The structures of claudins and claudins bound to cCpE were critical to understanding their trans and cis assembly as well as CpE’s mechanisms for disrupting tight junction barriers by offering important new insights that generated many testable hypotheses. Whether CpE binding occurs outside of tight junctions before claudins integrate into polymerized strands, whether this occurs at the apical or basolateral compartments (or both), and whether CpE can directly and actively break claudin/claudin interactions at polymerized tight junctions remains to be established. Although the structures of claudin/cCpE complexes were unable to answer such questions unequivocally, they were nonetheless foundational for narrowing down the hypotheses that best explain decades worth of findings from biochemical, biophysical, and cell-based analyses.

## 7. Applications of CpE for Therapeutic Use

The use of claudins as diagnostic and prognostic biomarkers or therapeutic targets has gained traction over the last decade, especially for treatments of human cancers. A recent review by Li et al. highlights the current state and availability of potential clinically relevant claudin-targeting agents [88]. Claudins play roles in nearly all aspects of tumor development and are known to be tumor suppressors or promotors [89]. Recent studies have demonstrated the overexpression of claudins in many cancers, including pancreatic, uterine, breast, gastric, and ovarian cancer [88,90]. Because several of the claudin subtypes that are overexpressed in human cancers are also CpE receptors, CpE is being actively investigated for translational applications and therapies in cancer. These applications include the targeted destruction of claudin-expressing cancer cells by CpE and the in vivo visualization of claudin-expressing cancers using radio or fluorescently labeled CpE. This normally gut-specific interaction is being exploited to target claudin-expressing cancer cells for CpE-mediated cytotoxicity in these cells. Recent work has employed CpE to induce cytolysis in cancerous breast, ovarian, colon, prostate, and gastric cells, most of which express claudin-3 and claudin-4 [91,92,93,94,95,96,97,98,99]. Moreover, other novel approaches for cancer treatment include the use of non-cytotoxic cCpE fused to antitumor reagents for targeted drug delivery to claudin-expressing tumors. Tumor necrosis factor was fused to cCpE and shown to be efficiently delivered and more cytotoxic to ovarian cancer cells than un-fused tumor necrosis factor [100]. Moreover, gold nanoparticles conjugated to cCpE have been shown to bind claudin-expressing tumors and then be selectively destroyed via nanoparticle-mediated laser perforation [101]. Likewise, polysialic acid-based nanoparticles conjugated to cCpE then loaded with doxorubicin were shown to specifically target claudin-4-expressing pancreatic cancer and suppress tumor growth while exerting low toxicity to normal tissues [102]. Other uses include non-invasive imaging of claudin-4-overexpressing pancreatic cancers using radiolabeled or fluorescently labeled cCpE in conjunction with single-photon emission computerized tomography scanning or fluorescence reflectance imaging and fluorescence-mediated tomography [103,104]. Similar optical approaches using fluorescent labeled cCpEs have been used to detect claudin-3 and -4 in ovarian cancer [105]. The structural biology of claudins and claudin/cCpE complexes have helped enable these advances, and their insights will likely be utilized in the future to improve the binding affinity and selectivity of CpE-based therapeutics.

CpE-based cancer treatments and diagnostics would ideally work best at nanomolar concentrations so that they can specifically target cancers overexpressing high-affinity CpE receptors, such as those involving claudin-4, -6, -9, and -17. The reason for this is that the conservation of sequence and structure in human claudins allows many subtypes to bind and/or be cytotoxically damaged by CpE at low micromolar concentrations. We, and others, have demonstrated this previously with claudin-1, -3, -8, -14, and -19, which all bind to CpE with lower affinities than claudin-4. At ~1 µM concentrations, cells expressing these claudins are killed by CpE [52,69,71,81]. The reason for this is that some claudins will be CpE-bound even at concentrations below their K_D_. Thus, a 1 µM CpE-based therapeutic may lack specificity toward the high-affinity claudins and bind partially to numerous subtypes, limiting their potential use and increasing their unintended side effects. To overcome this bottleneck, a structure-guided design of mutant cCpE- and CpE-based therapeutics can be developed to improve the binding affinity to normally weaker receptors, increasing their targeted selectivity. This approach has already been employed. Piontek et al. recently generated a CpE mutant that showed claudin-1-dependent binding and cytotoxicity in thyroid and lung cancer cells [106]. Because thyroid and non-small-cell lung cancers highly express claudin-1 and/or -5—low-affinity CpE receptors—modified CpEs could be useful as treatments. However, because claudin-4 has a ~1000-fold higher affinity for CpE compared to claudin-5, it is still likely that modified CpEs engineered to bind claudin-5 will still possess appreciable binding capacity to high-affinity claudins, such as claudin-4. If claudin-4 and -5 are expressed at equivalent amounts in similar tissues, the effects may be off-target. As such, care in analyses is warranted. Equal prudence must be taken when interpreting results from non-human animals and translating them to humans. When comparing human claudin orthologs to their counterparts in the mouse model organism, although their sequences diverge only slightly, some orthologs are known to bind CpE with as great as 26-fold differences in affinity, as is the case for claudin-3 [69]. Because mouse claudin-3 is a high-affinity CpE receptor and human claudin-3 is a moderate-affinity receptor, results from mouse models of claudin-3-expressing cancers may appear more specific and, thus, cannot be used directly to predict outcomes in humans.

Other translational efforts that employ cCpE seek to modulate the leakiness of very highly regulated tight junctions, such as the blood-brain barrier (BBB). The BBB is the diffusion barrier that limits paracellular transport of molecules from the blood to the brain and is formed by an interdependent network of endothelial cells whose properties are precisely regulated by tight junctions [107,108]. The use of cCpE is under investigation to tune paracellular permeability of BBB tight junctions to deliver drugs to the brain, a notoriously challenging route [109]. Many traditional and budding strategies for modulating the BBB for drug delivery via opening of tight junctions have been reviewed recently by Han [110]. Of all the BBB claudins, claudin-5 is predominantly expressed in brain endothelia and, therefore, has been the most highly studied and targeted claudin for BBB modulation, although other subtypes have emerged as alternative targets [111,112]. Using in vitro models of the BBB and mutagenized cCpE, cCpE is being tested to determine if it can transiently open BBB tight junctions [113,114]. Utilizing a structure-based approach for modifying cCpE, Protze et al. created a Tyr306Trp and Ser313His mutant that enhanced the binding to claudin-5-expressing and tight junction-less HEK293 cells [113]. They suggested that the absence of a bulky hydrophobic residue at position 151 in claudin-5 and the presence of Asp149 and Thr151 hinder its binding to cCpE. Their cCpE double mutant serves as a “pocket filler” and compensates for these residues, resulting in an estimated binding of 33 nM [109]. This mutant decreased trans-endothelial electrical resistance (TEER) in a concentration-dependent and reversible manner in primary brain endothelial cells and cerebEND cells from various species and increased the permeability of carboxyfluorescein in claudin-5-expressing brain endothelial cells without a total breakdown of tight junction integrity [114]. It was noteworthy that the observation that the cCpE double mutant bound stronger to claudin-1 but more weakly to claudin-3, -4, -6 and -9 compared to wild type cCpE [113]. Similar findings were observed when cCpE was modified to make interactions more specific for claudin-5 with Asn218Gln, Tyr306Trp, and Ser313His mutations [114]. Liao et al. built on this work by showing that the Tyr306Trp and Ser313His double cCpE mutant modulated BBB permeability both in vitro and in vivo, where passage of rhodamine B-dextran leaked through zebrafish cerebral vascular cells [115].

In addition to BBB applications, the use of cCpE for the intentional opening of tight junction barriers is also being explored in the small intestine and epidermis, where cCpE was found to enhance the transmucosal absorption of dextran and opened the epidermal barrier in reconstructed human epidermis [91,116]. Takahashi et al. has screened a cCpE mutant library and isolated a broad claudin-binding mutant called m19, which contains the following mutations: Ser304Ala, Ser305Pro, Ser307Arg, Asn309His, and Ser313His [117]. They showed that the m19 cCpE could bind claudin-1, -2, -4, and -5 as well as increase paracellular transport in epithelial cells and enhance jejunal absorption. Applications that employ cCpE to modulate or transiently open tight junctions, it’s given function, have the potential to provide novel and non-invasive systems for drug delivery through normally restrictive tight junction barriers. Such applications could be employed to treat the wide array of disorders and pathologies that involve CpE receptor claudins in tissue- or cell-specific manners [11,24]. Structure-guided efforts to redesign cCpE for new, more targeted functions, coupled with high-throughput analyses, also have the potential to rapidly progress these endeavors.

## 8. Outstanding Questions and Future Directions

Many outstanding questions still impede a deep mechanistic understanding of CpE-based disruption of tight junction barriers. What is the function of ECH1, and is it a conserved feature of claudin structure? Currently, only one claudin structure has been determined in the absence of cCpE, that of mCLDN-15. To understand the effects of cCpE binding to the claudin structure, new structures must be determined of claudins in their apo forms, a technically challenging task because cCpE facilitates crystallization. All hypotheses regarding the cCpE-based disabling of claudin cis assemblies are based on comparisons to the mCLDN-15 structure that solely revealed ECH1. No other ECH1 has been observed experimentally. In fact, when searching for claudins in the AlphaFold Protein Structure Database, these artificial intelligence-derived models do not predict a full ECH1 for any ortholog [118]. More structures of apo claudins need to be determined to validate the mCLDN-15 result, to train the artificial intelligence algorithms to predict more accurate models, and to refine hypotheses specific to the cCpE-based disabling of cis assemblies. Moreover, what are the structures of the CpE pre-pore and β-pore? The structures of a “small” claudin/CpE complex as well as the claudin-bound CpE pre-pore and cytotoxic β-pore require experimental structure determination. No mechanistic understanding of the process of cytotoxic CpE β-pore formation can occur without them. Although CpE oligomers can be crystallized, they are not functionally relevant units. Similarly, while “small” complexes form in vitro, they do not assemble into functional pre-pore or β-pore complexes in the absence of membranes. What accessory molecules are required to signal the formation of the cytotoxic CpE β-pore? Most of these questions can only be answered with structural biology approaches. Enthusiastically, the structures that result from these pursuits will surely provide unanticipated avenues of inquiry for new and more complex questions to be posed. Structural data accompanied by biophysical binding measurements will enable a quantitative understanding of claudin/CpE function. These measurements are useful for classifying the hierarchy of claudin receptors for CpE, for instance, as well as for interpreting static in crystallo structures in the context of dynamic interactions in solution. These structural and biophysical approaches are especially important to aid the design of CpE-based therapeutics to quantitatively determine their affinities and selectivities for their intended claudin(s)—improving their efficacies.

Although structural and biophysical approaches can provide important insights, in the absence of in vivo understanding, they are incapable of holistically elucidating CpE functionalities. What structural biology has helped to usher, however, is a focusing in on a select few plausible mechanisms for CpE disruption of tight junctions, as outlined in Section 6. To definitively explain CpE’s mode of action in the various tissues of the gut, we need to determine if CpE acts exclusively at apical or sub-apical compartments. To do so will also clarify whether pools of claudins localize to these domains. The structural and functional diversity of digestive tissues and cells during homeostasis and development imposes a challenge. In the colon, apical claudins are accessible to CpE, and perhaps this is true elsewhere at given times for CpE to exploit? If CpE works sub-apically, how does it gain access to these compartments through the tight junction barrier? Are other Clostridium perfringens molecules accessories to CpE action? Further, we need to decipher if CpE works indirectly to disrupt tight junctions through cell-death-inducing mechanisms, or directly by disabling claudin assemblies. The former supposes a cytotoxicity-induced breakdown of tight junctions through cell signaling cascades. The latter supposes direct disabling and/or disordering of claudin/claudin interactions. Does CpE *disable* claudin assemblies, i.e., CpE binding to claudins alters their structure in a way that no longer enables cis or trans assembly? Disabling may occur via a “sequestering” mechanism that changes the equilibrium of claudin polymerization into tight junction strands. Alternatively, does CpE *disorder* claudin assemblies, i.e., claudins are cis or trans assembled at tight junctions and then their intermolecular interactions are broken upon binding by CpE? Disordering may then be classified as an active and direct process for breaking down tight junction barriers. Understanding such CpE in vivo functions, in conjunction with the abovementioned structural insights, will unravel a deep mechanistic understanding of the CpE-based disruption of tight junction barriers. One that will advance the design of specific, better targeted, and more effective cCpE- and CpE-based therapeutics to diagnose and destroy cancer cells and modulate impassible tight junction barriers, providing innovative and potentially life-saving treatments for these and a range of other complex diseases.

## 9. Concluding Remarks

In less than eight years, nine structures of five unique claudin subtypes have been determined by X-ray crystallographic methods, whereas no structural information existed for the 15+ years after the discovery of claudins as the structural backbone of tight junctions. Eight of the nine structures attribute their successful structural determination to cCpE. Not only have these structures illuminated the claudin fold and an individual claudin’s function but together with the knowledge gained by CpE structures they all have contributed to a rapid, comprehensive, and mechanistic understanding of the structural and functional biology of CpE and its claudin receptors at sub-molecular levels. The insights provided by this structural information have synthesized several decades worth of functional data into a cohesive but still incomplete understanding of tight junction barriers. It is expected that continued progress will be made by answering currently unanswered questions of which structural biology will likely play an indelible role. The role of structural biology is already apparent in the advancements of cCpE- and CpE-based diagnostics and therapeutics. Hence, future innovations will require research groups from multidisciplinary fields to be interdependent to enable treatments for claudin-linked diseases and improve human health. Even in the absence of translational applications, CpE will continue to be an increasingly powerful tool to examine the structure and function of tight junctions.

## Figures and Tables

**Figure 3 cells-11-00903-f003:**
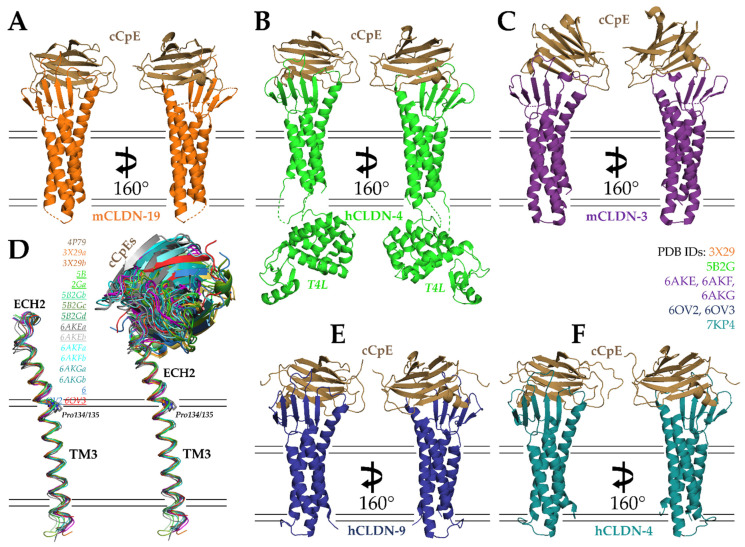
Structures of cCpE in complex with claudins. (**A**) cCpE bound to mCLDN-19 (PDB ID: 3X29). (**B**) cCpE bound to T4L-hCLDN-4 (PDB ID: 5B2G). (**C**) cCpE bound to mCLDN-3 (PDB ID: 6AKE). (**D**) Structural overlays of claudins bound to cCpE and focusing on TM3 and ECH2, represented as ribbons and colored according to their PDB ID. Lowercase letters after the PDB ID depict chains in the asymmetric unit. The PDBs used for comparison are shown in the figure, with Pro134-containing claudins underlined. cCpE is depicted as a cartoon and colored identically to its corresponding claudin. Movements of cCpE appear to induce TM3/ECH2 alterations. (**E**) cCpE bound to hCLDN-9 (PDB ID: 6OV2). (**F**) cCpE bound to hCLDN-4 (PDB ID: 7KP4). For (**A**–**C**,**E**,**F**), cCpE (brown) and claudins (various) are represented as cartoons and colored. mCLDN-19 (orange), T4L-hCLDN-4 (green), mCLDN-3 (purple), hCLDN-9 (blue), and hCLDN-4 (teal).

**Figure 4 cells-11-00903-f004:**
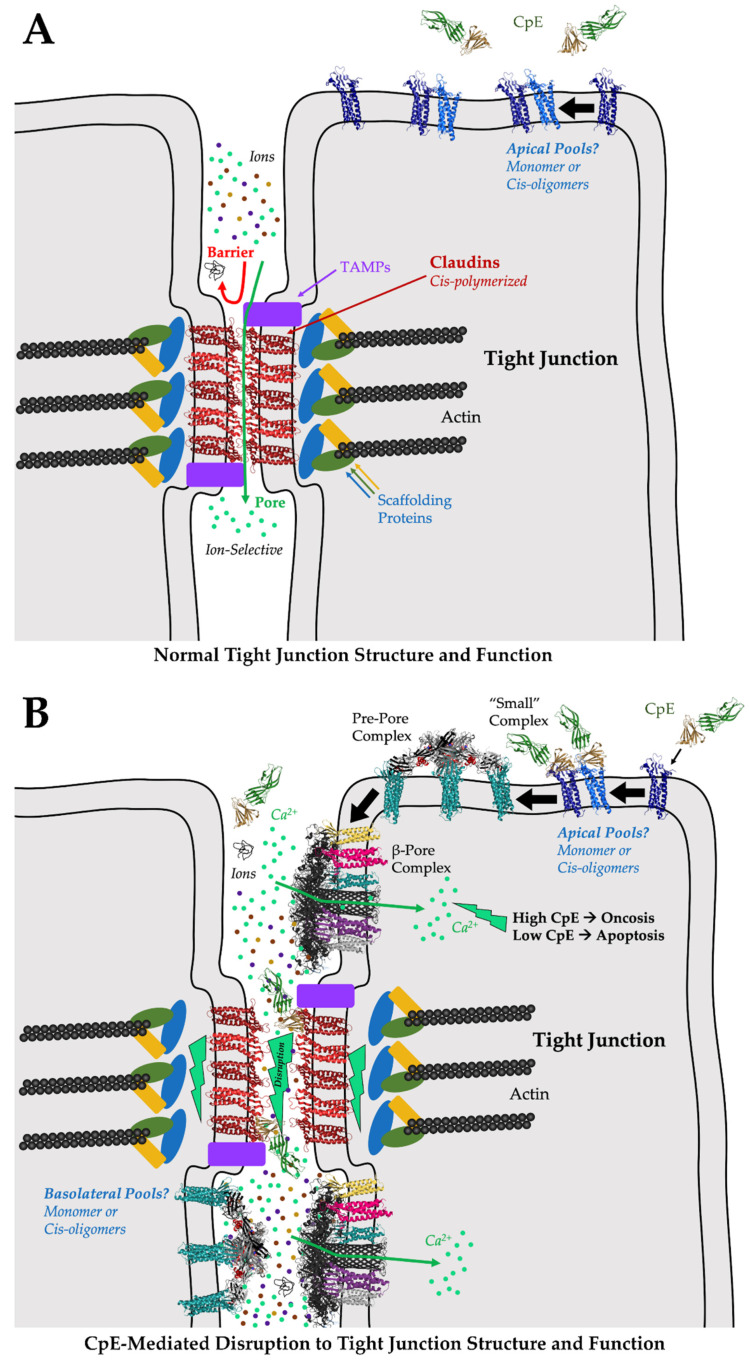
Model of CpE-mediated disruption of tight junction structure/function. (**A**) Normally, the cis-polymerization and trans assemblies of claudins on adjacent cells form tight junctions at intercellular “kissing points”. Claudin assembly yields barriers to large and small molecules. Depending on the constituent claudins, pores are formed that are ion selective. The selective molecular properties of tight junctions, imparted by claudins, further imparts tissue-specific functions to epithelia. (**B**) During *Clostridium perfringens* infection, CpE is produced. CpE may target and bind to apical claudin monomers or cis oligomers outside of tight junctions, forming “small” complexes. “Small” complex formation initiates a cascade of events that generates a pre-pore complex composed of claudin-bound CpE hexamers; ultimately leading to the formation of a β-pore complex comprised of the same molecules. The CpE β-pore is structurally distinct in that CpE’s nCpE domain rearranges, forming a membrane-penetrating β-barrel pore. The pore allows the intracellular influx of Ca^2+^, initiating either oncosis or apoptosis depending on CpE concentrations. The cell death signals disassemble and degrade cis-polymerized and trans-assembled claudins at tight junctions, disrupting the tight junction barrier and allowing unfettered molecular leakage and further CpE-mediated insults to the basolateral compartments. Gut homeostasis is disrupted by these or similar processes. The model of the CpE β-pore presented here was generated by superimposing the aerolysin pore (PDB ID: 5JZT) onto the CpE portions of the model for the claudin-bound pre-pore complex (Figure 2C).

**Table 1 cells-11-00903-t001:** Summary of claudin/cCpE structures from the PDB.

Claudin	Modification	cCpE	PDB	Resolution (Å)	A.U.	Ref.
mCLDN-19	Truncation of the C-term and depalmitoylation of membrane-proximal cysteines (104, 183, and 184) with alanine.	cCpE^Ser313Ala^	3X29	3.7 Å	Anti-parallel dimer (1:1)	[74]
hCLDN-4	N-terminal fusion with T4L and truncation of the C-term.	cCpE	5B2G	3.5 Å	Anti-parallel tetramer (1:1)	[77]
mCLDN-3	Truncation of the C-term and depalmitoylation of membrane-proximal cysteines (103, 106 181, and 182) with alanine.	cCpE^Ser313Ala^	6AKE 6AKG (Pro134Gly) 6AKF (Pro134Ala)	3.6 Å 4.3 Å 3.9 Å	Anti-parallel dimer (1:1)	[79]
hCLDN-9	Full-length and wild type	cCpE	6OV2 (closed) 6OV3 (open)	3.2 Å 3.3 Å	Monomer (1:1)	[81]
hCLDN-4	Full-length and wild type	cCpE	7KP4	3.4 Å	Monomer (1:1)	[69]

## Data Availability

All data for protein structures referenced in the review are freely available from the Protein Data Bank at https://www.rcsb.org/ (accessed on 6 February 2022).

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
