# Peer review of "Disruption of Claudin-Made Tight Junction Barriers by Clostridium perfringens Enterotoxin: Insights from Structural Biology"

_cells, 2022, doi:10.3390/cells11050903_

Round 1

Reviewer 1 Report

The manuscript entitled “Disruption of Claudin-Made Tight Junction Barriers by Clostridium perfringens Enterotoxin: Insights from Structural Biology” by Ogbu et al. provides a very comprehensive, highly interesting and informative overview about the interaction of C. perfringens enterotoxin (CpE) with claudins and potential mechanisms of cCpE- and CpE-mediated tight junction alterations with a focus on the insights provided by structural biology. This review article will be of very high value for the field of structural biology and (patho-)physiology of tight junctions and tissue barriers. However, there are several concerns that have to be addressed before the manuscript can be accepted for publication.

Comments:

Abstract

  1. Abstract “CpE binds claudins at tight junctions in the gut, disabling their assembly and thus their barrier function.”

As mentioned by the authors in the manuscript, it is not clear whether CpE binds claudins at or outside of tight junctions and if binding to claudins directly disables their assembly. Thus, this should by rephrased.

  1. p2 “which showed that all claudins share a conserved WGLWCC motif”

Not all claudin share this motif, for instance, mammalian claudin-10b and claudin-15 contain WNLWCC, thus,  for instance, “WG/NLWCC motif” is more appropriate.

  1. p2 “The first experimental structure of a claudin, that from mouse claudin-15 (mCLDN-15) came in 2015 from Fujiyoshi and co-workers”

It came in 2014, Protein Data Bank (PDB) ID: 4P79) [28].

  1. p5 „It is believed that CpE binding to claudins occurs outside of tight junctions to surface-exposed claudins, as application of CpE to only basolateral compartments affect tight junction barrier integrity [39,57].”

In these two studies cCpE but not CpE was used for analysis of tight junction barrier integrity.

  1. p6 “The peptide bound in a groove in cCpE between β8 and β9 as predicted previously [59]. However, the peptide’s orientation in the claudin-binding pocket was opposite to those from prior modelling studies [66]. Crystal structures determined later of cCpE in complex with claudins would reveal that the claudin-2 peptide was indeed bound backward”

The orientation of the claudin ECS2 loop in the corresponding binding groove of cCpE was successfully predicted using in silico and in vitro data [66] and verified by all subsequent non-peptide claudin/cCpE complex structures.  The sentences should be rephrased or complemented to make this more clear.

  1. p9 “The structure showed where Lys65 and other paracellular pore-lining residues responsible for anion selectivity in the inner ear reside; explained how a Phe35Leu in hCLDN-9 could cause deafness through destabilization of cation barriers; and where Val32 and Val35 reside and how they may be used by hepatitis C virus for cell entry [80,81]”

These sentences give the impression, it was established knowledge that claudin-9 forms a anion pore and that the analysis of the claudin-9 structure revealed that and how Lys65 and other proposed pore-lining residues are involved in a anion selectivity of claudin-9 channels.

To my knowledge, regarding the topics mentioned, at least the claudin-9 topic is largely unclear. Thus, it has to be clarified in detail or removed.

7.:

 p16 “, some orthologs are known to bind CpE with as great as 26-fold differences in affinity, as is the case for claudin-3 [69].

p9 “.Surprisingly, we found that human claudin-3 (hCLDN-3) had 100-fold and human claudin-1 (hCLDN-1) had 380-fold lower affinity for cCpE compared to hCLDN-4”

“The discovery of the cCpE-binding motif explained results from Katahira et al. in 1997 that showed claudin-3 had lower receptor capacity for CpE than claudin-4; and linked the previously conflicting theories that proposed that either electrostatics or NPLVA153 driven interactions govern cCpE binding to claudins [27,43,57,70,71].”

“Moreover, we showed that hCLDN-1 and -3 are not viable receptors for CpE at pathophysiologically relevant concentrations, which range from 3 to 350 nM [71]. Therefore, we hypothesized, the native receptor for CpE in humans is hCLDN-4, while in mice it is mCLDN-3 and potentially -8.”

Analysis of binding of125I-CPE to human claudin-3 or human claudin-4 in a native mammalian membrane environment (L929 cells) resulted in KDs of 21,7 nM and 12,6 nM, respectively (hCldn4 Katahira et al. 1997, J Biol Chem 272, 26652-26658). This small difference (< 2-fold) contrasts the 100-fold difference reported for claudins solubilized from non-mammalian cells. This discrepancy should be discussed. 

It is not clear in which respect the mentioned 12-amino acid “cCpE-binding motif” “discovered” in the mentioned study (69) explains results from Katahira et al. in 1997 and other labs better than previous work (57, 72, 66, 75). These studies linked already hydrophobic and electrostatic contributions in cCpE-claudin interactions. Furthermore, electrostatic interaction mediated by Asn148/Asn149 in claudin-3/claudin-4 - that strongly contributes to binding to cCpE - was shown previously by biochemical as well as structural studies (57, 66, 72, 14, 75, 77). However, if I understood it correctly, Asn148/Asn149 was excluded from the 12-amino acid “cCpE-binding motif”. Please clarify and rephrase “The discovery of the cCpE-binding motif explained results from Katahira et al. in 1997 that showed claudin-3 had lower receptor capacity for CpE than claudin-4; and linked the previously conflicting theories that proposed that either electrostatics or NPLVA153 driven interactions govern cCpE binding to claudins [27,43,57,70,71].”

  1. p6 „Finally, these works provided a foundation for elucidating the amino acid determinants of CpE recognition of claudins, which, through biochemical analyses, had yielded contradictory findings up to that point. Structures of claudin/enterotoxin complexes would be essential to clarify these ambiguous functional results.”

Which contradictory findings do the authors refer to? See also comment 7.

  1. p11 “this work helped to clarify these complexities and surmised that the cis1 interface facilitates barrier or pore formation by either electrostatic attraction or repulsion between claudin ECS1 domains, respectively.”

This seems to be not the case. Hempel et  al. (32) suggested that barrier versus pore formation could be at least partly mediated in the center of the channels by electrostatic attraction or repulsion,  respectively, and at the entrance of the channels by shifted ECS2–ECS2 trans-interactions.  Please clarify and rephrase.

  1. p12 “Sonoda et al. speculated that this “sequestering” mechanism could function by CpE occupying non-polymerized pools of claudins and preventing their integration into polymerized strands if tight junctions are indeed maintained by such an equilibrium [39]. Their observation that CpE only affected tight junction barrier integrity when applied to basolateral and not apical compartments appears to validate this mechanism”

This was done with cCpE and sequestering was suggested for cCpE. CpE might open tight junctions by Ca2+ Influx-induced alterations. 

  1. “Disabling of claudin/claudin trans interactions through sequestering may very well be the only direct mechanism of CpE-induced disruption of tight junctions.”

This might be the case for cCpE. What is the indication that this may not only potentially  but “very well” (very likely?) be the case for CpE?  

  1. p12 “To validate this, further experiments are required to understand how or whether CpE gains access to basal compartments; or if non-polymerized pools of claudins are synthesized apically, polymerized, and are then integrated into tight junction strands.”

p18 “To definitively explain CpE’s mode of action, we need to determine if CpE acts at apical or sub-apical compartments—to do so will also clarify whether pools of claudins localize to these domains.”

Further experiments in this direction suggesting that claudins can appear on the apical surface due to pathophysiological or transient tight junction leaks were reported and could be mentioned/discussed here (Eichner et al., 2017. J Infect Dis. 217:147-157).

  1. p13“ Apo hCLDN-4 dimers would form intermolecular interactions between Tyr67, Leu71 and Leu77; while cCpE-bound dimers would not, due to unwinding of the ECH1 helix”

How can intermolecular interactions between Tyr67, Leu71 and Leu77 be formed in hCLDN-4 dimers?

“These two structures of mCLDN-19 and hCLDN-4 bound to cCpE provided a new mechanism for cCpE-induced disruption to tight junctions, which involves its disabling of cis interactions.”

In the corresponding publications, a new mechanism for cCpE-induced disruption to tight junctions  involving disabling of cis interactions was proposed. However, no experimental in vitro support for this idea was provided. Please rephrase.  

Figure 4: How was the CpE pore structure generated?

Author Response

We  thank reviewer 1 for their thorough reading of the review and their thoughtful comments. Rather than describe point-by-point in all instances what was changed, we will say that we have implemented all of reviewer 1 comments into the revised manuscript, and these changes can be easily and quickly viewed via track changes. For some points that need further description, we will address these below.

Point 6: we have added appropriate references that established claudin-9 as an anion pore-forming claudins and re-wrote that part of the paragraph to better elaborate on this.

Point 7: Regarding the more modest differences between claudin-3 and -4 binding to CpE from Katahira—we now list several reasons for this potential difference as compared to our results, as well as reference two papers that showed greater disparity in receptor binding using similar methods to Katahira. Note that mouse claudin-19, which is not considered a receptor, binds cCpE with 70-fold weaker affinity than human claudin-4 (Shinoda et al, Saitoh et al). We have data (unpublished) that shows human claudin-19 affinity is ~50-fold lower than human claudin-4 so comparatively these results are consistent and demonstrate weak binding by human claudin-3 and justify its classification.

Point 7: Regarding cCpE-binding motif Asn148/149— in the review we tried to summarize the findings of this paper rather than extend its discussion. Therefore, we feel it may be out of the scope of this review to elaborate in great detail, for instance, on why Asn148/149 is not included in the motif. The original publication is intended for this. In short though and to answer the question, the motif is unique because it pinpoints and predicts residues present in high-affinity CpE binders (not necessarily those involved in interactions). Asn148/149, for instance, although involved in cCpE binding and present in many claudins is not a distinguishing residue of high-affinity receptors. Mouse claudin-15 and human claudin-19, for instance, have an Asn148/149 but are not considered CpE receptors. In the paper we show human claudin-3, which has this Asn, does not bind with high affinity. The motif is a distinguishing feature of high-affinity claudins, and this is why some residues are not contained within it. The motif does not pinpoint the only or most important residues involved in CpE binding. I hope this explanation clarifies things. For other comments in this regard, we altered the text to reflect the contributions of other findings. The comment “These studies linked already hydrophobic and electrostatic contributions in cCpE-claudin interactions.” may be a result of difference of interpretation. In Kimura et al JBC 2010 regarding their proof of electrostatic interactions being the driving force for toxin binds, the authors use a paragraph of the discussion to critique Winkler et als JBC 2009 findings regarding the NPLVA motif. Here they note the “unreliability” of their approach and note the contradictory findings in sensitivity found between the two papers. Our interpretation is thus at that historical time point (~2010), it was very much still a debate in the field whether electrostatics or (not and) the NPLVA motif drove interactions, despite Winkler discussing the importance of the Tyr pit and NPLVA motif together. Structures helped to settle this debate, we would argue. The cCpE-binding motif does help to bring these ideas into focus. But the electrostatics proposed by Kimura are now known via structure (some determined at low pH) to not be the major driver although electrostatics do occur but outside the area they proposed. Structure shows that shape complementarity imposed by the NPLVA motif rather than intermolecular forces seem to influence this association the most. This is not to say that the amino acids described to be involved by Kimura and Winkler and others that are outside of the motif are not important—multifaceted factors influence the entirety of this recognition—but some are more critical, like Leu153.

Point 8: Regarding contradictory findings—as this is introduced at the end of section 4 we felt it best to elaborate in the beginning of section 5 to reduce redundancy. We therefore changed the word contradictory to ambiguous in section 4 and expounded with examples on this in the beginning of section 5.

Thank you again for your great read and comments of the review and we hope we addressed them adequetely.

Reviewer 2 Report

The manuscript entitled "Disruption of Claudin-Made Tight Junction Barriers by Clostridium perfringens Enterotoxin: Insights from Structural Biology" is a quality review, which provides information of the structural biology for claudins in complex with cCpE. In the current work, the authors sought to deliver important insights on CpE-based disruption of tight junction barriers in the field of cancer, and may contribute to the advancements of cCpE- and CpE-based diagnostics and therapeutics, particularly in the field of cancer.

The review is well planned, well developed, and the narratives are clear, profound and convincing, and raise no doubts in my mind.

Only a few questions/comments remain to be clarified:

CpE is being actively investigated for translational applications and therapies in the field of cancer, given that several claudin subtypes are overexpressed in human cancers are also CpE receptors. However, there was little illustration or investigations for the role and function for CpE binding claudins at tight junctions in other diseases/disorders except for cancers. Could the authors provide relevant literatures for elucidation of the tissue-specific functions associated with diverse disorders/diseases for discussion?

Proof reading is needed, such as improper space in sentences.

Author Response

We thank reviewer 2 for their thorough review of the review. To address their question—if I understand the comment, the reviewer would like us to highlight or reference what other diseases and tissue-specific functions claudins that bind CpE are involved in. This is an interesting point and we have added a sentence to the last paragraph of section 7 that includes references describing the tissue-specific diseases associated with CpE receptor claudins. We feel this sentence expands the potential application of these technologies. We cannot expand on this topic much more because, to our knowledge, these applications are more poorly developed.

Thank you again for your help in making this review better.

Round 2

Reviewer 1 Report

The authors have sufficiently addressed most of the concerns